# Pharmacokinetic Markers of Clinical Outcomes in Severe Mental Illness: A Systematic Review

**DOI:** 10.3390/ijms24054776

**Published:** 2023-03-01

**Authors:** Pasquale Paribello, Mirko Manchia, Federica Pinna, Ulker Isayeva, Alessio Squassina, Claudia Pisanu, Lorenzo Balderi, Martina Contu, Marco Pinna, Bernardo Carpiniello

**Affiliations:** 1Section of Psychiatry, Department of Medical Sciences and Public Health, University of Cagliari, 09124 Cagliari, Italy; 2Unit of Clinical Psychiatry, University Hospital Agency of Cagliari, 09124 Cagliari, Italy; 3Department of Pharmacology, Dalhousie University, Halifax, NS B3H 4R2, Canada; 4Section of Neuroscience and Clinical Pharmacology, Department of Biomedical Science, University of Cagliari, 09042 Cagliari, Italy; 5Lucio Bini Mood Disorders Centers, 09128 Cagliari, Italy

**Keywords:** pharmacogenomics, severe mental illness, systematic review, precision psychiatry, prediction

## Abstract

The term severe mental illness (SMI) encompasses those psychiatric disorders exerting the highest clinical burden and socio-economic impact on the affected individuals and their communities. Pharmacogenomic (PGx) approaches hold great promise in personalizing treatment selection and clinical outcomes, possibly reducing the burden of SMI. Here, we sought to review the literature in the field, focusing on PGx testing and particularly on pharmacokinetic markers. We performed a systematic review on PUBMED/Medline, Web of Science, and Scopus. The last search was performed on the 17 September 2022, and further augmented with a comprehensive pearl-growing strategy. In total, 1979 records were screened, and after duplicate removal, 587 unique records were screened by at least 2 independent reviewers. Ultimately, forty-two articles were included in the qualitative analysis, eleven randomized controlled trials and thirty-one nonrandomized studies. The observed lack of standardization in PGx tests, population selection, and tested outcomes limit the overall interpretation of the available evidence. A growing body of evidence suggests that PGx testing might be cost-effective in specific settings and may modestly improve clinical outcomes. More efforts need to be directed toward improving PGx standardization, knowledge for all stakeholders, and clinical practice guidelines for screening recommendations.

## 1. Introduction

Mental and substance use disorders are leading causes of disability on a global level [1], with a significant portion of this burden deriving from severe mental illnesses (SMIs) [2]. Collectively, SMI represents an ill-defined category which has been inconsistently reported in the literature in the field [3] but that, as a bare minimum, comprises conditions such as schizophrenia (SCZ), bipolar disorder (BD), and major depressive disorder (MDD) [2,3]. Among individuals affected by SMI, life expectancy has been reported to be reduced by 20 years among males and up to 15 years among females [4]. In the past, this gap in life expectancy was frequently attributed to suicide risk. However, over the years, it has been increasingly evident how cardiovascular and infectious disorders also represent significant causes of death in this population [4,5,6]. The toll associated with SMI is not limited to the affected individuals but extends to their relatives and communities [7]. Carers of individuals affected by SMI may indeed report lower employment levels, and social and economic difficulties with higher levels of food insecurities [8] and expenditures related to care [9]. Individuals affected by SMI represent a severely underserved population, despite significant advancement in their management. For example, only 41% of individuals affected by MDD may receive treatment at minimal standard of care [10]. Even for the minority of individuals receiving treatment, finding the most effective therapeutic option could be challenging for healthcare providers and service users. In fact, even when the most updated protocols are employed, the treatment choice is based on a “trial-and-error” approach, which ultimately may result in frequent treatment failures and significant healthcare costs [11,12]. Numerous factors should be considered when discussing the basic underpinnings for the observed heterogeneity in treatment response (HTR), such as the nosological classification systems used for the diagnoses [13,14,15,16], age of onset, co-morbidities, and clinical course. These factors likely represent a source of HTR intrinsic to the current standards of practice [17]. Notwithstanding the previously mentioned limitations, this framework has produced most of the evidence for treatments (either pharmacological or psychotherapy) in psychiatry since clinical trials testing the efficacy and tolerability of a particular intervention have indeed selected study patients based on a categorical nosological system [17]. Waiting for the development of more accurate diagnostic tools [18], one possible way to address HTR would be to tailor treatments to the individuals identified through the use of the current nosological classification systems by matching the right treatment to the right patient [19,20,21,22]. In this setting, a growing body of evidence suggests that pharmacogenomics (PGx) may represent a useful tool for enabling personalized treatments. PGx is the research area dedicated to evaluating how multiple genetic variations may interact and influence the metabolism and action of a particular pharmacological treatment [23]. With very few notable exceptions (e. g., lithium salts, gabapentin), nearly all medications currently employed for the treatment of psychiatric disorders are metabolized in the liver. The major metabolic reactions involved in the process are oxidation (phase I) and conjugation (phase II). Genetic variations for transporters expressed at different locations, such as the brain, gut, and liver, can also influence the pharmacokinetic profile of the different compounds employed in treatment, but their clinical impact has not been established [23]. The metabolic system that has been most extensively studied is represented by cytochrome P450 (CYP450), comprising 57 genes and 58 pseudogenes [24]. The two isoenzymes of CYP450 most extensively studied for psychiatric treatments are CYP2D6 and CYP2C19, as there is significant evidence that these two can significantly influence psychotropic metabolism [24,25], with CYP2D6 being involved in the metabolism of almost half of the most prescribed psychotropics [25]. For a long period of time, it has been known that single-nucleotide polymorphisms (SNPs) could be associated with differential gene expression profiles and that these, in turn, could be studied to help estimate the risk of developing adverse effects or to quantify treatment response to a particular medication in a subgroup of individuals [22]. Allelic variants of CYP genes are indicated with an asterisk (*), genotypes are then coded based on their projected metabolic activity, and the corresponding phenotypes are typically subdivided into Rapid, Normal, Ultrarapid, Intermediate, and Poor Metabolizer [23]. Genes supposedly associated with the postulated mechanism of action at the biochemical, cellular, and physiological level are instead associated with the pharmacodynamic of a particular compound. In psychiatry, attention has been focused on possible allelic of genes involved in neurotransmitters’ receptors, signal transmission, gene transcription, or protein folding, among others [23]. Gene variations in human leukocyte antigens or in proteins regulating immune mechanisms have also been the subject of research in the area and have yielded guidance in the projected risk of developing adverse reaction upon exposure to certain compounds [23,26]. To improve the accessibility to treatment-informing guidance based on PGx, several scientific bodies have developed clinical practice guidelines with the most significant being summarized on easily accessible platforms such as PharmGKB [26]. In theory, PGx holds great promise in terms of improving personalization of treatments as it would aid clinicians in streamlining the pharmacological treatment selection based on the expected efficacy and tolerability for the different available pharmacological treatments [11]. However, in psychiatry the clinical application of this tool has lagged behind due to concerns regarding its efficacy and lack of knowledge on interpreting its results in a sizeable portion of healthcare providers. In the present study, we performed a systematic review of the literature in the field probing the use of PGx for SMI, specifically reporting on pharmacokinetic markers of treatment response, as defined by the authors. Importantly, we applied for the first time a transdiagnostic approach to explore whether we could be able to identify PGx markers associated with similar patterns of response across disorders. The main objective of this project is reviewing the existing evidence for pharmacokinetic markers in predicting pharmacological treatment response in individuals affected by SMI, focusing on the comparison with the usual standard of care when available.

## 2. Materials and Methods

A double-blind systematic review was performed on Scopus, PubMed, and Web of Science according to the Preferred Reporting Items for Systematic Reviews and Meta-Analyses (PRISMA [27]). In this project, we considered including articles published in English probing the association of PGx tests with pharmacological treatment outcomes for SMI (i.e., BD, MDD, SCZ) and reporting on pharmacokinetic markers. We defined treatment outcomes as a response to the practiced treatment regimen, as reported by the authors. Accepted study designs included: (1) open-label trials, (2) randomized controlled trials, (3) cross-sectional studies, (4) retrospective cohort studies, (5) prospective cohort studies, and (6) studies recruiting human subjects ≥ 18 years old and assessing treatment outcomes as defined by the study authors. We excluded: (1) meta-analyses, (2) systematic reviews, (3) case reports, (4) case series, (5) letters to the editor, and (6) editorials. No time restriction was applied based on the year of publication. Pharmacodynamic markers and studies assessing the safety or tolerability profile of pharmacological treatments were excluded. The following search strategy was employed (“pharmacogenomic” OR “pharmacogenomics” OR “pharmacogenetics” OR “pharmacogenetic”) AND (“signature” OR “biomarkers” OR “marker” OR “determinants”) AND (“severe mental illness” OR “severe mental disorders” OR “schizophrenia” OR “psychosis” OR “schizoaffective disorder*” OR “bipolar disorder *” OR “major depressive disorder *”). Two reviewers independently screened the records identified through the primary search strategy. With the objective of reviewing the existing evidence for pharmacokinetic markers in predicting pharmacological treatment response in individuals affected by SMI, we focused on extracting the following data from the included studies: (1) study design, (2) sample composition, (3) main objective, (4) inclusion and (5) exclusion criteria, (6) country where the selected study was performed, and (7) reported outcomes pertinent to our project. The qualitative data extraction was performed independently by two authors (P.P.; L.B.) and whenever a discrepancy was found a third senior author was involved to reach a consensus. Rayyan, a semi-automated tool, was employed to facilitate the screening process [28]. The primary search was further augmented using a comprehensive pearl-growing strategy. ROB 2 [29] was employed for the assessment of bias for randomized controlled trials by two independent raters. Again, discrepancies were solved through discussion and, if needed, with a third author’s judgement. The last search was performed on the 17 September 2022. All tables are available in interactive mode on GitHub (https://github.com/claudiapis/tables_pharmacokinetic_markers, accessed on 25 February 2023). Further, the main input set is available on GitHub (https://github.com/pasqualeparibell/Pharmacokinetic-markers-of-clinical-outcomes-in-severe-mental-illness-a-systematic-review.---source/tree/main, accessed on 25 February 2023).

## 3. Results

### 3.1. Search Results and Bias Assessment

The selected search strategy resulted in the identification of 1975 records. After duplicate removal, 1456 records were assessed through an abstract and title screening, leading, in turn, to the identification of 587 records. Among them, 42 papers were selected for the qualitative analysis, summarized in 3 different tables dedicated to (1) SCZ (Table 1 and Table 2) MDD (Table 2 and Table 3) and BD (Table 3). A complete description of the selection process is reported in the PRISMA flow diagram, in Figure 1.

A total of 13 studies originated in the USA and 14 in Asia. The remaining studies were carried out mainly in European countries. Among the included studies, eleven were randomized controlled trials (RCTs), with 10 recruiting individuals affected by MDD and only one focusing on individuals affected by SCZ [30]. One RCT on MDD recruited a mixed sample of individuals affected by MDD and/or anxiety, but no description of the anxiety disorder was included [31]. Only three of the included studies reported on individuals affected by BD, with one of the three including a heterogeneous population comprising MDD, BD, and post-traumatic stress disorder (PTSD) [32]. Overall, the risk of bias of the included RCTs appears limited, save for three studies, judged at high risk of bias [33,34,35]. Figure 2 summarizes the bias assessment for the included RCTs according to ROB 2.

### 3.2. PGx Outcomes

Reported outcomes included service use reduction, symptom change from baseline, and rates of remission or response to treatment. In line with the literature in the field, there was significant heterogeneity in scales employed to report the symptom changes. The description of the sample composition of the included studies also appears inconsistent, with the vast majority providing the gender composition, age range, or the average age of the recruited sample. A discrete heterogeneity also emerged regarding the employed inclusion or exclusion criteria, even considering the heterogeneity of the analyzed diagnostic categories. Numerous different alleles and genotypes have been assessed, but no specific efficacy pattern emerged for a particular marker across the various studies. A relatively limited number of studies [32,34,36,37,39,41,42,43,44] reported on the results of combinatorial PGx testing, introducing a further layer of complexity in the interpretation of the results for the included studies.

#### 3.2.1. Schizophrenia

Seventeen papers reported on studies comprising individuals affected by SCZ, with only one randomized controlled trial (RCT) [30]. Among them, ten papers reported on the possible association between CYP2D6 and treatment outcomes as described by the authors [30,45,46,47,48,49,50,51,52], four papers described the association between ABCB1 genotypes and treatment outcomes with three out of four reporting a positive association [47,52,53,54]. Overall, a significant heterogeneity of assessed outcomes is apparent. Two papers used retention in the treatment of antipsychotics (AP) as the primary outcome [30,55]. As for symptom severity assessment, seven papers used the Positive and Negative Symptoms Scale (PANSS) score as a primary outcome measure, either focusing on total percent change or changes in some of its subscales [45,49,50,52,55,56,57], whilst seven papers used the Brief Psychiatric Rating Scale (BPRS) percent change [49,51,53,54,58,59,60]. Seven out of a total of seventeen papers reporting on SCZ described a positive association between PGx markers of efficacy with treatment outcomes [45,48,49,52,53,54,58]. One study [52] assessed the association of pharmacodynamic together with pharmacokinetic markers of efficacy. An additional paper [60] focused on the association of PGx tests with the change in BPRS-defined cognitive symptoms of SCZ. These results are summarized in Table 1.

**Table 1 ijms-24-04776-t001:** Selected papers reporting on individuals affected by SCZ.

Author, Year	Study Design	Sample Size	Sample Characteristics	Diagnostic Category	Main Outcomes Reported	Inclusion Criteria	Exclusion Criteria	Country	Main Results
Almoguera et al., 2013 [45]	Open-label study	75	M 41, F 34; -<40 y.o. (*n* = 25) -40–59 y.o. (*n* = 39) - ≥ 60 y.o. (*n* = 11)	SCZ (DSM-IV)	Association of PGx test and RIS treatment outcomes	-Admitted to an acute inpatient unit	N/A	SPN	*CYP2D6* poor metabolism appeared to be associated with greater T-PANSS improvement
Gregoor et al., 2013 [46]	Case–control study	528	-Cases *n* =222 (M 151, F 71) -Controls *n* = 306 (M 161, F 145)	SCZ (DSM-IV)	Association of *CYP2D6* phenotypes and probability of switching to CLZ	-Psychotic disorder -Received two AP trials of at least one month each	N/A	NET	No statistically significant association between *CYP2D6* phenotypes and outcomes
Grossman et al., 2008 [55]	CATIE Cohort	708	See Suppl. Material	SCZ (DSM-IV)	Association of 25 functional variants in metabolizing enzymes with treatment outcomes	-18–65 y.o.	-Intolerance or no response to treatments -Pregnancy, breastfeeding	USA	No association with treatment efficacy
Jürgens et al., 2020 [30]	RCT	669	-GGT *n* = 95 (mean age: 42 y.o.) -SCM *n* = 94 (mean age: 40 y.o.) -CG *n* = 101 (mean age: 42 y.o.)	SCZ	Association of *CYP2D6* or *CYP2C19* phenotypes and 1-year AP treatment outcome	-No previous PGx testing -≥ 18 y.o.	-Not adherent to the treatment protocol	DEN	No association between the tested phenotypes and treatment outcomes
Jovanović et al., 2010 [47]	Open-label study	83	M 17, F 66 (mean age 30.3 ± 8.1 y.o.)	SCZ (DSM-IV)	Association of *CYP2D6 (*3,*4,*6), ABCB1 (G2677T/A and C3435T)* genotypes with 8-week RIS treatment outcomes	-FEP -No prior AP exposure -Oral RIS -Diazepam and biperiden only two medications allowed	N/A	CRO	No statistically significant association between the tested polymorphisms and symptom change
Kaur et al., 2017 [48]	Open-label study	443	M 262, F 157 (mean age: 31.3 ± 9.5 y.o.)	SCZ (DSM-IV)	Association of *CYP2D6* phenotype with 12- week RIS treatment outcomes (response T-PANSS ≥ 50% from baseline)	-18–55 y.o. -Caregiver that could monitor treatment adherence	-SUD other than tobacco -MR -LAI -Metabolic syndrome -Severe medical or surgical co-morbidity	IND	*CYP2D6*4* polymorphism frequency differed significantly in terms of T-PANSS change when drop-outs were excluded from the analysis
Lesche et al., 2019 [56]	Retrospective cohort study	66	M 48, F 18 (mean age: 40 ± 10 y.o.)	SCZ (DSM-IV)	Association of *CYP1A2, CYP2D6, CYP2C19* phenotypes with CLZ treatment outcomes (T-PANSS score)	-On CLZ treatment -18–65 y.o.	N/A	AUT	*CYP1A2, CYP2D6, CYP2C19* activity score impact appears limited compared to nongenetic factors
Lin et al., 2006 [58]	Open-label study	41	M 33, 8 F (mean age 35.6 ± 8.8 y.o.)	SCZ (DSM-IV)	Association of three PGP polymorphisms and 6-week OLA treatment outcomes (% change BPRS)	-18–65 y.o. -BPRS ≥ 42	-SUD -Unstable medical illness	USA	For *ABCB1 3435T* carriers, OLA plasma levels correlated with % BPRS change
Lu et al., 2021 [49]	Open-label study	76	M 38, F 38–mean age: 45 y.o.	SCZ	Association of *CYP2D6* phenotypes and 8-week RIS treatment	-No AP exposure for 1 year and recent hospital admission	-ECT in the 3 months prior to enrollment -Medical illnesses	CHN	Significant changes in PANSS score between *CYP2D6* phenotypes
Müller et al., 2012 [50]	Retrospective cohort study	35 SCZ, 39 OCD	N/A	SCZ, OCD (DSM-IV)	Association of pharmacokinetic markers and treatment response	N/A	N/A	CAN	No statistically significant association between *CYP2D6* phenotypes and treatment outcomes
Nikisch et al., 2011 [57]	Open-label study	22	M 14, 8 F (age range: 22–49 y.o.)	SCZ (DSM-IV)	Association of clinical response with *ABCB1* with 4-week quetiapine treatment outcomes (PANSS change)	-PANSS ≥ 60 -CGI ≥ 2 -Started on QUE	-Other DSM-IV Axis I diagnosis -CLZ -LAI -Severe somatic conditions	GER	Noncarriers of the *3435TT* genotype showed greater changes in the PANSS score
Nozawa et al., 2008 [51]	Open-label study	51	M 34, F 17 (mean age: 32.6 ± 9.6 y.o.)	SCZ (DSM-IV)	Association functional polymorphisms of *UGT1A4, CYP1A2,* and *CYP2D6* with OLA treatment outcomes	N/A	N/A	JPN	No association between the tested genotypes and BPRS change
Rajkumar et al., 2012 [59]	Retrospective cohort study	101	CLZ duration 4–174 months (73 M, 28 F; mean age: 35.4 ± 9.4 y.o.)	SCZ (DSM-IV)	*CYP1A2* SNP and treatment response	-TRS -Stable dose of clozapine for at least 12 weeks -South Indian ethnicity	-Neurological illnesses -Intellectual disability -Sensory impairment precluding the assessment	IND	No association with clozapine treatment response
Vijayan et al., 2012 [53]	Case–control study	192	Dravidian (responders: *n* = 130 (68%), nonresponders *n* = 62 (32%))	SCZ (DSM-IV)	Association of *ABCB1* gene polymorphisms with treatment response after 1 year AP	N/A	-SAD -Neurological or general medical condition that may precipitate psychotic symptoms	IND	Homozygous genotypes of *rs1045642* and *rs2032582* were associated with a better response
Xing et al., 2006 [54]	Open-label study	130	Han Chinese (45 M, 85 F; mean age 36.27 ± 11.18 y.o.)	SCZ (DSM-IV)	Association of nine polymorphisms of *ABCB1* gene with % BPRS improvement on RIS therapy	N/A	-No physical complications or other psychiatric co-morbidity, TRS -No previous exposure to 2nd GEN-AP	CHN	Individuals with *C1236T TT* genotype *(rs1128503)* presented higher % improvements
Xu et al., 2016 [52]	Open-label study	995	Han Chinese	SCZ (DSM-IV)	Association of 77 single-nucleotide polymorphisms of 25 candidate genes and treatment response	N/A	-Physical co-morbidity -SUD -TRS -No previous AP exposure	CHN	Several associations emerged with various genes
Yasui-Furukori et al., 2006 [60]	Open-label study	33	Inpatients (mean age 37.3 ± 12.8 y.o.)	SCZ (DSM-IV)	Association of *MDR1* gene polymorphisms and clinical response to BPD (BPRS score change)	-BPRS ≥ 18 -No psychotropics four weeks before enrolment	-Psychiatric co-morbidity -Epilepsy -AUD, SUD -Significant physical or neurological disorders	JPN	-No association of symptom improvement with the *C3435T* genotypes

Abbreviations: AD—Antidepressant; 2nd GEN-AP—second generation antipsychotic; AP—Antipsychotic; AUD—Alcohol use disorder; AUT—Austria; BPD—Bromperidol; BPRS—Brief Psychiatric Rating Scale; CAN—Canada; CG—Control group; CGI—Clinical Global Impression; CHL—Chloropromazine; CHN—China; CLZ—Clozapine; CRO—Croatia; DEN—Denmark; DSM-IV—Diagnostic and Statistical Manual of Mental Disorders IV Edition; F—Female; FEP—First episode psychosis; GGT—Gene-guided treatment; GER—Germany; IND—India; JPN—Japan; LAI—Long-acting injectables; M—Male; MR—Mental retardation; MS—Mood stabilizer; N/A—Not available; *n*-PANSS—Negative subscale–Positive and Negative Symptoms Scale; NET—Netherlands; OCD—Obsessive compulsive disorder; OLA—Olanzapine; QUE—Quetiapine; RIS—Risperidone; SAD—Substance abuse disorder; SAPS—Scale for the Assessment of Positive Symptoms; SCA—Schizoaffective; SCZ—Schizophrenia; SCM—Structured clinical monitoring; SPN—Spain; SUD—Substance use disorder; P-PANSS—Positive Subscale–Positive and Negative Symptoms Scale; PANSS—Positive and Negative Symptoms Scale; PER—Perphenazine; T-PANSS—Total Positive and Negative Symptoms Scale; TRS—Treatment-resistant schizophrenia; USA—United States of America; y.o.—Years old; ZIP—Ziprasidone.

#### 3.2.2. Major Depressive Disorder

Twenty-three of the included studies focused on individuals affected by MDD [31,32,33,34,35,36,37,38,39,40,41,42,43,61,62,63,64,65,66,67,68,69,70], and ten of them were RCTs [31,32,34,35,36,37,39,40,43,68]. Seven papers specifically reported on the association of CYP2D6 polymorphisms and treatment outcomes as defined by the authors [33,40,64,65,69,70]. Four papers [38,61,63,66] focused on the association between ABCB1 genotypes/alleles and treatment outcomes. Sixteen studies employed the Hamilton Depression Rating Scale (HDRS) as the primary outcome measure; seven of them defined remission as HDRS ≤ 7 [31,35,36,39,42,63,67], whilst a single paper defined remission as HDRS ≤ 10 [70]. One paper employed the Structured Interview Guide for the Hamilton Depression Rating Scale (SIGH-D17) as the primary outcome measure [68], and three papers the mean HDRS change [38,41,43,61,64,65,66]. Other symptom rating scales employed as the primary outcome measure included the Quick Inventory of Depression Scale—Self Report (QIDS-SR) [40], Patient Global Impression of Improvement (PGI-I) [34], and the Patient Health Questionnaire-9 (PHQ-9) [37]. Two additional papers reported on the association between PGx testing and hospital stay duration [61,69]. One study recruited a mixed population of MDD and anxiety disorder (unclear diagnosis of the anxiety disorder) [31], and another one recruited individuals affected by MDD, BD, or PTSD [32]. These findings are illustrated in Table 2.

**Table 2 ijms-24-04776-t002:** Selected papers reporting on individuals affected by MDD.

Author, Year	Study Design	Sample Size	Sample Characteristics	Diagnostic Category	Main Outcomes Reported	Inclusion Criteria	Exclusion Criteria	Country	Main Results
Altar et al., 2015 [41]	Secondary analysis—open-label trial	334	-81 La Crosse -18 Hamm Clinic -20 Pine Rest -119 TAU -96 chart review	MDD	Combinatorial PGx test of four *CYP450* enzymes, *SLC6A4*, and *HTR2A* (mean change HDRS score)	-18–75 y.o. -HDRS ≥ 14 -Failed one medication trial	-SCZ, BD I, SCA	USA	The combinatorial PGx test discriminated between poor and good treatment outcomes
Bradley et al., 2018 [31]	RCT	685	-GGT *n* = 352 (M 95, F 257, mean age: 47.8 ± 14.5 -TAU *n* = 333 (M 92, F 241, mean age: 47.3 ± 15.2)	MDD and anxiety (DSM-5, unclear anxiety diagnosis)	Association of the NeuroIDgenetix test with 12-week treatment outcomes (remission HDRS ≤ 7, response HDRS change ≥ 50%)	-19–87 y.o. -Undergoing a new trial of AD (either as first or subsequent treatment trials)	-BD, SCZ, PD -Traumatic brain injury -Chronic kidney disease stages 4–5 -Malabsorption -Pregnancy -Abnormal hepatic functioning -High risk of suicide	USA	GGT group presented a higher remission and response rate for depression
Breitenstein et al., 2014 [61]	Open-label study	116	-GGT *n* = 58 M 26, F 32 (mean age: 48.5 ± 15.1) -Control *n* = 58-M 26, F 32 (mean age: 46.5 ± 14.6)	MDD	Association of *ABCB1* polymorphism testing and AD treatment outcomes (hospital stay, mean HDRS change)	-Admission to the involved hospital and *ABCB1* testing	N/A	GER	*ABCB1* test was associated with lower HDRS scores; *rs2032583, rs2235015,* and dose increase were associated with shorter hospital stays
Calabrò et al., 2022 [62]	Retrospective cohort study	540	Mean age: 52.2 ± 14.2	MDD (DSM-IV)	Association of *CYP2C19* phenotypes and treatment outcomes (response MDRS ≥ 50% change and MDRS < 22–TRD ≥ ineffective two trials of 4-week duration)	-MDRS 22 after adequate, 4-week AD trial	-SUD active previous six months	ITA	PM presented higher response and remission
Dong et al., 2009 [63]	Open-label study	272	Mean age: 38 ± 10 y.o.	MDD (DSM-IV)	Association of five SNPs of *ABCB1* gene with 8-week DES or FLX therapy (remission HDRS < 8)	-HDRS ≥ 18	-Active medical illness -High suicide risk -Pregnancy or breastfeeding -BDZ or AD treatment in the two weeks prior to enrollment -Illicit drug use in the three months before enrollment -Current involvement in psychotherapy -No Mexican American heritage	USA	No association between treatment outcomes and the tested *ABCB1* SNPs
Forester et al., 2020 [42]	Post hoc analysis—GUIDED cohort	206	M 56, F 150 (mean age: 49.4 ± 4.4 y.o.)	MDD (DSM-IV)	Association of combinatorial PGx testing and treatment outcomes with 24 weeks’ AD treatment among ≥ 65 y.o. (response HDRS ≥ 50% change from baseline, remission HDRS ≤ 7)	-≥18 y.o. -Inadequate response to AD treatment of ≥6 weeks’ duration or intolerability -QIDS-C16 and QIDS-SR ≥11 -Follow the protocol	-High suicide risk -BD I or II -Delirium, dementia, SCZ, or psychotic disorders -Psychotic features -Inpatient -Prior PGx test -Pregnancy or breastfeeding -ECT, DBS, or TMS -Gastric bypass -Unstable medical illness	USA	Higher remission and response rates at eight weeks of treatment
Greden et al., 2019 [36]	RCT	1167	-GGT 717—M 219, F 498 -TAU 618—M 192, F 489	MDD	Association of PGx testing (Genesight) with 24-week treatment outcomes (remission HDRS ≤ 7, QIDS-C16 ≤ 5, PHQ-9 ≤ 5)	-≥18 y.o. -QIDS-C16 and QIDS-SR16 ≥ 11 -At least one medication trial with inadequate response	-High suicide risk -Psychiatric or cognitive co-morbidity -HDRS ≤ 14	USA	GGT did not improve symptom severity change but improved remission and response rates
Gressier et al., 2014 [64]	Open-label study	87	-UM *n*= 11 -EM *n* = 141 -PM *n* = 21	MDD (DSM-IV)	Association of 4 weeks of flexible AD treatment and *CYP2D6* genotypes (% HDRS change and CGI-I).	-≥18 y.o. -HDRS ≥ 18	-Psychotic disorder -Brain disorders or unstable physical illness -SUD	FRA	No association between the studied phenotypes and treatment outcomes. Ultrarapid carriers taking CYP2D6 inhibitors had lower AD responses compared with the other genotypes
Han et al., 2013 [33]	Open-label study	94	-PP *n* = 28 (mean age: 48.2 ± 17.7) -PS *n* = 38 (mean age: 44.9 ± 15.2) -SS *n* = 28 (mean age: 44.8 ± 15.5)	MDD (DSM-IV)	Association of *CYP2D6 P34S* with 12-week ESC treatment outcomes (response HDRS ≥ 50% change from baseline)	-HDRS ≥ 18	-SCZ, SCZ, psychotic features in the past six months, BD, dementia -Personal or family history of AUD or SUD	SKR	*P allele* associated with better ESC treatment outcome as compared with the others
Jeon et al., 2009 [65]	Open-label trial	153	M 38, F 115	MDD (DSM-IV)	Association of *CYP2D6 P34S* polymorphisms and 12-week mirtazapine treatment outcomes (remission—unspecified criteria, HDRS and CGI-S % change)	-HDRS ≥ 18 -≥ 18 y.o.	-SCA, SCZ, AUD, SUD, dementia -Personal or family history of SUD -Use of BDZ or MS	SKR	*S allele* was associated with smaller changes in HDRS and CGI-S scores
Kato et al., 2008 [66]	Open-label study	68	N/A	MDD (DSM-IV)	Association of three *ABCB1* polymorphisms with 6-week PAR treatment outcomes (HDRS % change from baseline)	N/A	-SUD other than tobacco -Unstable medical illness -Pregnancy -ECT in the six months prior to enrollment	JPN	The *ABCB1 G2677T/A* genotype appeared to be associated with symptom changes
Lin et al., 2010 [67]	Open-label study	241	Han Chinese (mean age: 41 y.o.)	MDD (DSM-IV)	Association of *CYP1A2* genetic polymorphisms with 8-week PAR treatment outcomes (HDRS ≤ 7, HAM-A ≤ 17, CGI-S ≤ 2)	-HDRS ≥ 14 -No previous PAR exposure	-SCZ, SCA, SUD, dementia	TWN	Three SNPs appeared to be associated with MDD remission
McCarthy et al., 2021 [32]	RCT	182	-GGT *n* = 75, (mean age: 52.5 ± 1.5 y.o.) -TAU *n* = 74 (mean age: 50.3 ± 1.6 y.o.)	MDD, BD, PTSD	Association of PGx markers and pharmacological treatment outcomes (CGI change with AD, AP, MS treatment)	-Failure of one or more first-line medications (AD or MS)	N/A	USA	No statistically significant difference in GGT vs. TAU response
Oslin et al., 2022 [37]	RCT	1944	-GGT (*n* = 966, M 737, F 229, mean age: 48 y.o.), -TAU (*n* = 978, M 716, F 262, mean age: 47 y.o.)	MDD	Efficacy of GGT in improving outcomes for MDD treatment (remission PHQ-9 ≤ 5)	-Receiving care at VAMC -18 to 80 y.o. -MDD with at least one previous episode -New trial of AD monotherapy	-SUD -BD -Psychosis -BOR, ASO -AP, buprenorphine or naltrexone augmentation -No bank account for payments	USA	Small nonpersistent improvement in remission rate for GGT as compared with TAU
Singh et al., 2015 [39]	RCT	148	-GGT *n* = 74 (M 31, F 43; mean age: 44.2) -TAU *n* = 74 (M 28, F 46; mean age: 44.3)	MDD (DSM-5)	Association of PGx with 12-week treatment outcomes between GGT and TAU (HDRS ≤7)	-HDRS < 18	-Psychotic disorders -BD -SUD -PD	AUS	Higher remission rates in the GGT group as compared with TAU
Pérez et al., 2017 [34]	RCT	316	M 115, F 201 -GGT (*n* = 155, mean age: 51.7 ± 12.0 y.o.) -TAU (*n* = 161, mean age: 50.7 ± 13.1 y.o.)	MDD (DSM-IV)	Association of PGx panel (Neuropharmagen^®^), with treatment outcomes (PGI-I ≤ 2 after 12-week AD treatment)	-CGI-S ≥ 4 -MDD -Starting a new treatment trial or medication switch	-No MDD diagnosis -Pregnancy -Breastfeeding -Treatment with strong 2D6 inhibitors (i. e. quinidine, cinacalcet, and/or terbinafine)	SPN	Higher treatment response in GGT as compared to TAU at 12 weeks
Perlis et al., 2010 [38]	Secondary analysis	250	M 92, F 158 (mean age: 44.2 ± 12.6 y.o.)	MDD (DSM-IV)	Association of polymorphisms of the *PDE1A, PDE1C, PDE6A, PDE11A, ABCB1, GRIK4, SLC6A4, OPRM1* genes with 7-week DUL treatment (HDRS score change)	-HDRS ≥ 15 allowed co-morbid GAD	-OCD	USA	Polymorphisms in *PDE1A, PDE1C, PDE6A, PDE11A, ABCB1, GRIK4, SLC6A4,* and *OPRM1* genes showed no statistically significant associations with duloxetine treatment response
Perlis et al., 2020 [68]	RCT	304	-GGT *n* = 151 (M 44, F 107; mean age: 47.8 ± 12.3) -TAU *n* = 153 (M 42, F 111; mean age: 47.6 ± 12.0)	MDD (DSM-IV)	Association of PGx test (Genecept Assay version 2.0) with 8-week AD treatment outcomes (change in SIGH-D-17 from baseline)	-SIGH-D-17 ≥ 18 -18–75 y.o. -Failure of at least one AD trial of adequate duration/dose	-Neurocognitive disorders, SCZ spectrum, personality disorders, BD, trauma disorders, PD	USA	No statistically significant difference in the SIGH-D-17 change between GGT and TAU
Ruano et al., 2013 [69]	Open-label study	149	Inpatients, M 58, F 91	MDD (DSM-IV)	Association of *CYP2D6* metabolism rate and hospitalization length (CIT 34.9%, QUE 31.5%, RIS 29.5, TRA 25.5%, VNL 18.8%, BUP 17.4%, SER 12.8%, FLX 12.8%, MRT 8.7%, ARI 4.7%, OLA 3.4%, TCAs 3.4%, ZIP 3.4%, PAR 3.4%)	-MDD requiring hospitalization	N/A	USA	Longer hospital stays among individuals with deficient *CYP2D6* metabolism
Tiwari et al., 2022 [43]	RCT	371	M 98, F 178; (mean age: 41.1 ± 14.1 y.o.)	MDD (DSM-IV)	Evaluated GGT vs. TAU after eight weeks of AD treatment (mean % HDRS change)	-≥18 y.o. -Inadequate response to at least one psychotropic included in the GGT test panel -QIDS-C16 score ≥11 at screening -QIDS-SR16 at screening and baseline	-Significant suicidal risk -Psychiatric or cognitive disorders, severe co-occurring psychiatric or cognitive disorders, and/or unstable or significant medical conditions	CAN	No difference in remission or response rate between GGT and TAU
Tsai et al., 2010 [70]	Open-label study	100	Han Chinese	MDD (DSM-IV-TR)	Association of *CYP450* polymorphisms (**4, *5*, and **10* on *CYP2D6, *2, *3*, and **17* on *CYP2C19*, and **18* on *CYP3A4*) and ESC response (remission HDRS ≤ 10 at 8 weeks)	-HDRS ≥14 -7-day washout AD	-Past failed trial on ESC	TWN	Intermediate *CYP2D6* metabolism is associated with ↑ rates of remission
van der Schans et al., 2019 [40]	RCT	106	M 40, F 66	MDD (DSM-IV)	Association of *CYP2D6* genetic variations and treatment outcomes with NOR or VNL (QIDS-SR score change)	-≥ 60 y.o.	-AD other than VNL or NOR -Medications that may interact with VNL or NOR -AST or ALT elevations -<30 mL/min GFR	NET	No significant differences between genotypes for depression severity
Winner et al., 2013 [35]	RCT	51	-GGT= M 8, F 18 (mean age: 50.6 ± 14.6) -TAU= M 2 F 23 (mean age: 47.8 ± 13.9)	MDD (DSM-IV)	Association of PGx testing (Genesight) with 12-week treatment outcomes (response HDRS ≥ 50 % change, remission HDRS ≤ 7)	-HDRS ≥ 14	-Requirement for inpatient treatment -ECT -SCZ, SCA, BD	USA	No statistically significant improvement for GGT as compared with TAU in remission

Abbreviations: AD—Antidepressant; ALT—Alanine transaminase; AP—Antipsychotic; ARI—Aripiprazole; ASO—Antisocial personality disorder; AST—Aspartate transaminase; AUD—Alcohol use disorder; AUS—Australia; BD—Bipolar disorder; BD I—Bipolar disorder type I; BD II—Bipolar disorder type II; BDZ—Benzodiazepines; BOR—Borderline personality disorder; BUP—Bupropion; CAN—Canada; CGI—Clinical Global Impression; CGI-I—Clinical Global Impression of Improvement; CGI-S—Clinical Global Impression of Severity; CIT—Citalopram; DBS—Deep brain stimulation; DES—Desipramine; DSM-IV—Diagnostic and Statistical Manual of Mental Disorders IV Edition; DSM–IV-TR—Diagnostic and Statistical Manual of Mental Disorders IV Edition Text Revision; DSM-5—Diagnostic and Statistical Manual of Mental Disorders 5th Edition; DUL—Duloxetine; ECT—Electroconvulsive Therapy; ESC—Escitalopram; EM—Extensive Metabolizer; F—Female; FLX—Fluoxetine; FRA—France; GAD—General anxiety disorder; GER—Germany; GFR—Glomerular filtration rate; GGT—Gene-guided treatment; HAM-A—Hamilton Anxiety Rating Scale; HDRS—Hamilton Depression Rating Scale; ITA—Italy; JPN—Japan; M—Male; MDD—Major depressive disorder; MDRS—Montgomery–Åsberg Depression Rating Scale; MS—Mood stabilizer; MRT—Mirtazapine; N/A—Not available; NET—Netherlands; NOR—Nortriptyline; OCD—Obsessive compulsive disorder; OLA—Olanzapine; PAR—Paroxetine; PD—Panic disorder; PGx—Pharmacogenomic; PHQ-9—Patient Health Questionnaire-9; PM—Poor Metabolizer; QUE—Quetiapine; QIDS-C16—Quick Inventory of Depressive Symptomatology—Clinician rated; QIDS-CR—Quick Inventory of Depressive Symptomatology—Clinician Rated; QIDS-SR—Quick Inventory of Depressive Symptomatology—Self Report; RCT—Randomized Controlled Trial; RIS—Risperidone; SCA—Schizoaffective disorder; SCZ—Schizophrenia; SER—Sertraline; SIGH-D-17—Structured Interview for Hamilton Depression Rating Scale—17 items; SKR—South Korea; SNP—Single-nucleotide polymorphism; SUD—Substance use disorder; TAU—Treatment as usual; TCA—Tricyclic antidepressants; TMS—Transcranial magnetic stimulation; TRA—Trazodone; TRD—Treatment-resistant depression; TWN—Taiwan; UM—Ultrarapid Metabolizer; USA—United States of America; VNL—Venlafaxine; y.o.—Years old; ZIP—Ziprasidone.

#### 3.2.3. Bipolar Disorder

Three papers reported on PGx’s association with clinical outcomes in individuals affected by BD. One paper [71] described the association between *CYP2D6* and symptom improvement as defined according to the Clinical Global Impression Efficacy Index (CGI-E). An additional paper [32] reported the association of CGI changes with PGx testing of a mixed population comprising BD, PTSD, and MDD. One paper [44] probed the potential cost savings associated with PGx-guided pharmacological therapy changes, focusing on emergency service access. These results are listed in Table 3.

**Table 3 ijms-24-04776-t003:** Selected papers reporting on individuals affected by BD.

Author, Year	Study Design	Sample Size	Sample Characteristics	Diagnostic Category	Main Outcomes Reported	Inclusion Criteria	Exclusion Criteria	Country	Main Results
Callegari et al., 2019 [44]	Prospective cohort study	30	Mean age: 48 ± 15 y.o.	BD (DSM-IV)	Association of PGx testing with emergency service access (cost associated with emergency service use)	-CGI-S ≥ 3 -Discordant therapy compared to what was suggested by PGx therapy in the 12 months prior to study start -Concordant therapy with PGx in the 12 months after the beginning of the study	N/A	ITA	Significant cost savings associated with switching to PGx-concordant testing
Huilei et al., 2020 [71]	Open-label study	200	-GGT—M 68, F 31 -TAU—M 70, F 30	BD (DSM-IV)	Association of *CYP2D6* with 12-week pharmacological treatment outcomes (CGI-E)	-HDRS ≥ 20	-Serious medical illness -≥2 failed treatment trials -Use of medications that might interact with practised treatment	CHN	GGT appeared to be associated with greater efficacy compared with TAU
McCarthy et al., 2021 [32]	RCT	182	-GGT *n* = 75, (mean age: 52.5 ± 1.5 y.o.) -TAU *n* = 74 (mean age: 50.3 ± 1.6 y.o.)	MDD, BD, PTSD	Association of PGx markers and pharmacological treatment outcomes (CGI change with AD, AP, MS treatment)	-Failure of one or more first-line medications (AD or MS)	N/A	USA	No statistically significant difference in GGT vs. TAU response

Abbreviations: AD—Antidepressant; AP—Antipsychotic; BD—Bipolar disorder; CGI-E—Clinical Global Impression Efficacy Index; CHN—China; DSM-IV—Diagnostic and Statistical Manual of Mental Disorders IV Edition; HDRS—Hamilton Depression Rating Scale; F—Female; GGT—Gene-guided treatment; ITA—Italy; M—Male; MDD—Major depressive disorder; MS—Mood stabilizer; N/A—Not available; PGx—Pharmacogenomic; PTSD—Post-traumatic stress disorder; RCT—Randomized controlled trial; TAU—Treatment as usual; USA—United States of America; y.o.—Years old.

## 4. Discussion

A growing amount of evidence points to the potential that PGx holds for treatment personalization in medicine [72,73,74], with notable examples of its applications in cardiology [75], oncology [76], pediatrics [77], and primary care [74], among others. With the right type of information support, PGx may further enhance the shared decision-making between service users and healthcare providers [78]. Great efforts have been invested in testing PGx’s efficacy in the pharmacological treatment selection for SMI, and our results seem to confirm our impression regarding its potential value. Meta-analyses of RCTs assessing the effectiveness of gene-guided treatment (GGT) versus treatment as usual (TAU) for MDD point to a modest but statistically significant benefit in terms of a higher remission rate for GGT as compared with TAU [79,80]. However, the clinical adoption of PGx testing in psychiatry appears somewhat delayed [11,23]. Over the years, several reasons have been proposed to explain this phenomenon. Among them, there are a relative lack of RCTs exploring PGx efficacy [11], a lack of knowledge on how to interpret its results by a sizeable portion of healthcare providers [11], inconsistencies in the guidance provided by different clinical practice guidelines [23], and an apparent lack of confidence in the overall value of PGx testing in clinical practice [11,81]. The results of our review seem to point to a significant heterogeneity in assessed outcomes and in the testing panels. Only three papers included in the present project reported on PGx testing in BD, with only one RCT [32]. Considering the current relatively limited number of papers dedicated to the topic, the evidence regarding PGx testing for treatment selection in BD appears particularly scarce. In our data synthesis, less than half of the total studies dedicated to SCZ reported a positive association between PGx and treatment outcomes, and among them three focused on ABCB1 polymorphisms and three additional papers reported on CYP2D6 polymorphisms. The only RCT included in this project and dedicated to assessing PGx testing in SCZ was negative [30]. At this stage, the evidence supporting the use of PGx testing alone to predict treatment outcomes in SCZ does not appear particularly poignant. Blood drug monitoring may represent an additional resource in guiding pharmacological treatment dosing, with clinical practice guidelines specifically dedicated to optimizing their use [82]. Arguably, PGx testing may be synergistically integrated with psychotropic blood monitoring to fully exploit these two different sources of information in optimizing the therapeutic and safety profile for each medication trial. Our study selection did not include any study employing combinatorial pharmacogenomic testing for predicting pharmacological treatment outcomes in SCZ. Fourteen of the twenty-three included studies focusing on MDD described a positive association between PGx testing and pharmacological treatment outcomes [31,33,34,37,39,41,42,61,62,65,66,67,69,70]. Five of the ten RCTs dedicated to MDD described a positive association for PGx testing and treatment outcomes [31,34,36,37,39], but considering the significant heterogeneity in the testing panels involved, no firm conclusion can be reasonably drawn from our results. Furthermore, a sizeable portion of the available evidence for PGx efficacy presents some financing biases, introducing additional complexity in the overall interpretation of the data [11]. Even pondering the results of the available meta-analyses may be a daunting task, as the proprietary nature of the tested algorithms employed in the involved studies hinders an accurate assessment of the relative impact of each approach [79,83]. Assessing the cost-effectiveness of PGx testing also needs careful consideration and individualized analyses. Commercial PGx costs vary significantly, and there might be differing reimbursement schemes depending on the geographic location with different corresponding healthcare systems and differing frequencies of actionable genotypes in the local population [11]. All these factors lead to the necessity of assessing cost-effectiveness profiles in the specific context where PGx testing should be employed [84,85]. The use of ethnicity as a guiding variable for treatment selection has been subjected to intensified scrutiny during recent decades. However, several clinical practice guidelines use the supposed ethnicity of origin as a possible element on which to base the decision on whether to perform PGx testing or not [86]. Ethnicity-based guidance for screening HLA-B*1502 among individuals of Asian ancestry prior to the use of carbamazepine, as an example, appears misguided and a potential source of confusion as HLA-B*1502 is nearly absent in South Korea and Japan [86]. Indeed, ethnicity represents a poor surrogate for the underlying biology. Therefore, such guidance should be abandoned in favor of more evidence-based, practical screening guidance [86]. Notwithstanding the previously mentioned limitations, a progressive cost reduction and a growing number of tested alleles may expand the number of individuals who may benefit from actionable treatment guidance. These factors, taken together, may increase PGx adoption in clinical practice [23]. Future efforts need to be devoted to improving the standardization for the tested algorithms and clinical practice guidelines, boosting educational programs on how to capitalize on PGx technologies in clinical care and assessing next-generation sequencing in PGx tests to address some of the lasting concerns surrounding PGx use [11,81,87,88].

### Limitations

The present paper focused on the association of pharmacokinetic markers, as the available evidence appears to be more solid as compared with pharmacodynamic markers. However, numerous papers have been published on the latter markers, and it would be worthwhile exploring the subject in future review projects. Indeed, the number of published studies on the field is far too great to be covered in a single paper. We did not include papers probing the eventual association between PGx testing and the safety or tolerability of pharmacological treatments. This might have led to the exclusion of a substantial part of the literature and of evidence supporting PGx testing in clinical practice. The search was limited to three databases and to articles written in English, wich could have also impacted on the extensiveness of our analyses. Finally, the lack of consistency in SMI’s clinical definition might have hindered our capacity of fully grasping the significance of PGx testing for predicting pharmacological treatment response in psychiatry.

## 5. Conclusions

A growing amount of evidence points to the potential that PGx testing holds for improving pharmacological treatment selection in psychiatry. PGx should be seen as an essential tool of an integrated approach which should take advantage of robust and standardized algorithms to help (but not solve) the decision-making process in terms of pharmacological interventions. Another neglected approach is represented by therapeutic drug monitoring, largely underutilized in SMI, but that could further boost the utility of PGx testing if adequately integrated with it. Future efforts will have to address lasting concerns surrounding the lack of standardization of the field and its practical implementation.

## Figures and Tables

**Figure 1 ijms-24-04776-f001:**
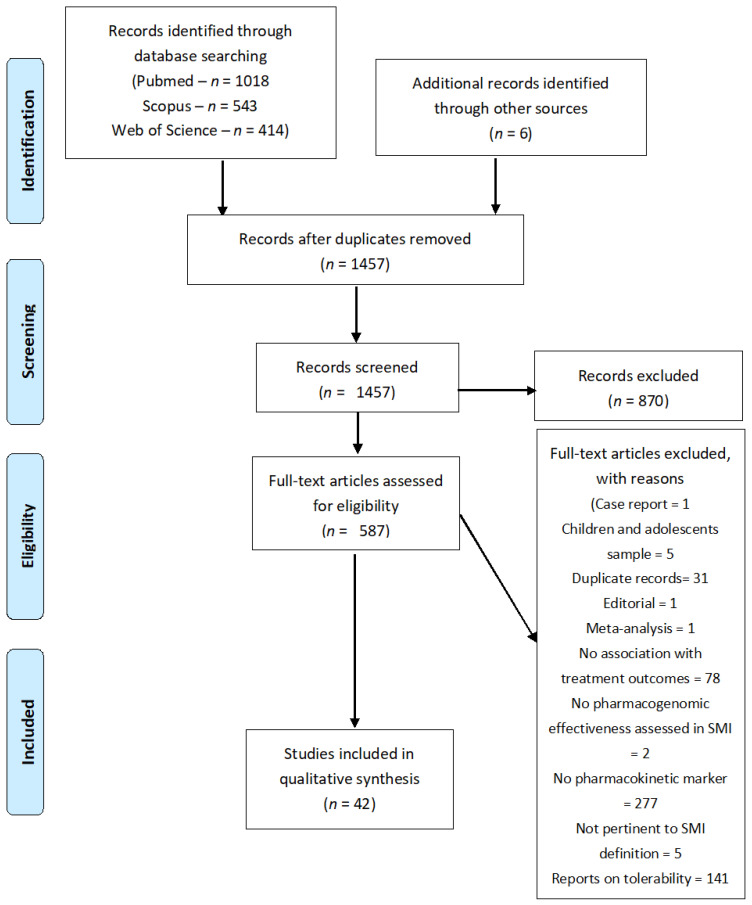
PRISMA 2009 flow diagram.

**Figure 2 ijms-24-04776-f002:**
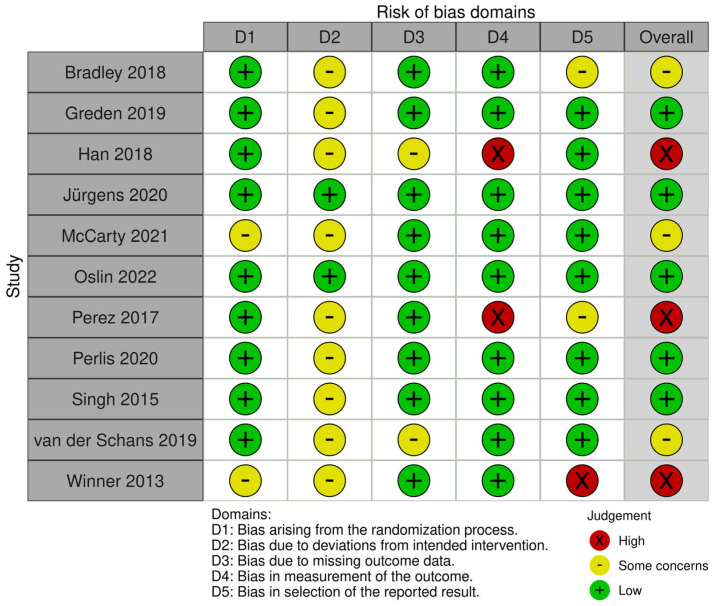
Traffic light plot for bias risk of included RCTs [30,31,33,34,35,36,37,38,39,40].

## Data Availability

No new data were produced as a result of this project.

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
