# Peer review of "Pharmacokinetic Markers of Clinical Outcomes in Severe Mental Illness: A Systematic Review"

_ijms, 2023, doi:10.3390/ijms24054776_

Round 1
Reviewer 1 Report
Thank you for the opportunity to review your paper. Clearly a thorough job has been done in preparation and review of the subject and creation of this review article.
The sentence in lines 47-49 is awkwardly worded, would consider revising.
In general, the introduction thoroughly introduces the topic and the difficulties in scientific study and classification, but may be a little lengthy. I would consider consolidating the introduction to focus on the specific topic of review. Specifically lines 52 – 77.
Line 163-word paper should be papers.
Table 1 though well organized and thorough is quite large, may be better served making a condensed version for publication and having the full table available as supplemental material.
Further it does not appear that the results of these papers presented in table 1 are summarized in the text of the results section of the paper, could consider doing this as away to summarize your data and discuss the findings in the discussion section.
Line 327-329 in conclusions, I don’t think you discussed this in the paper; Would consider supporting or omitting from conclusions.
Author Response
Thank you for the opportunity to review your paper. Clearly a thorough job has been done in preparation and review of the subject and creation of this review article.
R) We thank the reviewer for the positive assessment our work.
1) The sentence in lines 47-49 is awkwardly worded, would consider revising.
R) The sentence was revised as follows: “Individuals affected by SMI represent a severely underserved population, despite significant advancement in their management. For example, only 41% of individuals affected by MDD may receive treatment at minimal standard of care [10].”
2) In general, the introduction thoroughly introduces the topic and the difficulties in scientific study and classification, but may be a little lengthy. I would consider consolidating the introduction to focus on the specific topic of review. Specifically, lines 52 – 77.
R) We revised the specified lines as follows: “Numerous factors should be considered when discussing the basic underpinnings for the observed heterogeneity in treatment response (HTR), such as the nosological classification systems used for the diagnoses [13-16], age of onset, co-morbidities, and clinical course. These factors likely represents a source of HTR intrinsic to the current standards of practice [17]. Notwithstanding the previously mentioned limitations, this framework has produced most of the evidence for treatments (either pharmacological or psychotherapy) in psychiatry since clinical trials testing the efficacy and tolerability of a particular intervention have indeed selected study patients based on a categorical nosological system [17]. Waiting for the development of more accurate diagnostic tools [18], one possible way to address HTR would be to tailor treatments to the individuals identified through the use of the current nosological classification systems by matching the right treatment to the right patient [19-22].”
3) Line 163-word paper should be papers.
R) Corrected in the revised version.
4) Table 1 though well organized and thorough is quite large, may be better served making a condensed version for publication and having the full table available as supplemental material.
R) A condensed version of table 1 is now available in the draft and an integral version will be proposed as supplemental material. The line 138-140 have been edited to reflect the required changes as follows: “Among them, 42 papers were selected for the qualitative analysis, summarized in three different tables dedicated to 1) SCZ (Table 1), 2) MDD (Table 2), 3) and BD (Table 3), respectively (the integral omni-comprehensive version of table 1 is available as supplemental material).” Tables 1, 2, 3 are also available on GitHub at the following link https://github.com/claudiapis/tables_pharmacokinetic_markers
5) Further it does not appear that the results of these papers presented in table 1 are summarized in the text of the results section of the paper, could consider doing this as a way to summarize your data and discuss the findings in the discussion section.
R) We have expanded the section summarizing the findings of the identified papers. In addition, the following passage was added to the result section for Table 1: “Seven out of a total of seventeen papers reporting on SCZ described a positive association between PGx markers of efficacy with treatment outcomes [40, 44, 46, 47, 52-54]. One study [54] assessed the association of pharmacodynamic together with pharmaco-kinetic markers of efficacy. An additional paper [55] focused on the association of PGx tests with the change in BPRS-defined cognitive symptoms of SCZ.”
6) Line 327-329 in conclusions, I don’t think you discussed this in the paper; Would consider supporting or omitting from conclusions.
R) The discussion section has been edited to better reflect the required changes as follows: “Only three papers included in the present project reported on PGx testing in BD, with only one RCT [29]. Considering the current relatively limited number of papers dedicated to the topic, the evidence regarding PGx testing for treatment selection in BD appears particularly scarce. In our data synthesis, less than half of the total studies dedicated to SCZ reported a positive association between PGx and treatment outcomes, and among them three focused on ABCB1 polymorphisms and three additional papers reported on CYP2D6 polymorphisms. The only RCT included in this project dedicated to assessing PGx testing in SCZ was negative [27]. At this stage the evidence concerning PGx testing alone as a predictor of treatment outcome in SCZ does not appear particularly poignant. Blood drug monitoring may represent an additional resource in guiding pharmacological treatment dosing, with clinical practice guidelines specifically dedicated to optimize their use [79]. Arguably, PGx testing may be synergistically integrated with psychotropic blood monitoring to fully exploit these two different sources of information in assessing the therapeutic and safety profile for each medication trial. Moreover, our study selection did not include any study employing combinatorial pharmacogenomic testing for predicting pharmacological treatment outcomes in SCZ. Fourteen of the twenty-three included studies focusing on MDD described a positive association between PGx testing and pharmacological treatment outcomes [28, 30, 31, 33, 34, 36, 39, 56, 57, 60-62, 65, 66]. Five of the ten RCT dedicated to MDD described a positive association for PGx testing and treatment outcomes [28, 31, 35, 36, 39], but considering the significant heterogeneity in the testing panels involved, no firm conclusion can be reasonably drawn from our results.”
Reviewer 2 Report
I have completed my review of the manuscript titled “Pharmacokinetic markers of clinical outcomes in severe mental illness: a systematic review.”.
This topic is a very interesting and very important segment of this are.
The article demonstrates an adequate understanding of the relevant literature.
Despite of containing new and significant information, this study has seriously limitations.
The introduction is too long, quite scattered with data. It does not clearly introduce the reader to the purpose of this study or what is assumed. A clinical question is NOT defined according to the PICO Model. The necessary elements of the PICO should be explicit and clearly defined!
The goal is not clear. At the end of the introduction, the authors point out what they did, but they do not point out what they assume or the goal of all that. This is the key flaw in the introduction.
Why chapter Methods after Results? That's a key chapter!
inclusion criteria should be identifiable from, and match the PICO question. Specifically, studies selected for inclusion are not based on clearly defined criteria. So inclusion and exclusion criteria must be specific.
Why does the PRISMA flow diagram end with (n=N/A). That box is redundant.
The authors give a clear view of the results in the tables, but without providing any numerical results (only descriptions).
Also, the forest plot of effect sizes and the overall summary effect of studies were missed (here, I think of numerical results, not descriptive ones). There are no numerical results!!!???
Additionally, the forest plots, which need to summarize the findings, are missing. For example, forest plot estimates (e.g., mean) and confidence intervals (e.g., 95% CI) represented by whiskers for studies and/or multiple findings within a study in a horizontal orientation, are missed.
Specifically, the systematic review needs to be presented as the best form of evidence because it is positioned at the top of the hierarchy of evidence. Accordingly, the summary results (numbered results) are very important to see the main result and give the main conclusion of this study.
What are the limitations of this study??
The implications for practices or future research are not quite clear because a summary result from the forest plot is missing.
In view of the extensiveness of the research area and the extensiveness of the number of studies, the discussion is too short and deficient.
This systematic review does not give a clear conclusion on the clinical significance of the research question (which was not defined). Also, the heterogeneity of this systematic review is missing (in results), and that result should confirm whether the selected works are consistent with the research question, generate a concise conclusion to this study, and impact future studies. The conclusion does not clearly indicate the significance of improving pharmacological treatment selection in psychiatry.
Author Response
I have completed my review of the manuscript titled “Pharmacokinetic markers of clinical outcomes in severe mental illness: a systematic review.”.
This topic is a very interesting and very important segment of this are.
The article demonstrates an adequate understanding of the relevant literature.
Despite of containing new and significant information, this study has seriously limitations.
1)The introduction is too long, quite scattered with data. It does not clearly introduce the reader to the purpose of this study or what is assumed. A clinical question is NOT defined according to the PICO Model. The necessary elements of the PICO should be explicit and clearly defined! The goal is not clear. At the end of the introduction, the authors point out what they did, but they do not point out what they assume or the goal of all that. This is the key flaw in the introduction.
R: Thank you for your feedback. The introduction section has been condensed and the following passage has been added in rows 132-135:” The main objective of this project is reviewing the existing evidence for pharmacokinetic markers in predicting pharmacological treatment response in individuals affected by SMI, focusing on the comparison with the usual standard of care when available.”
2) Why chapter Methods after Results? That's a key chapter!
R: Thank you for your feedback. This is in accordance with the template for the International journal of molecular science, available at the following link https://www.mdpi.com/files/word-templates/ijms-template.dot.
3) Inclusion criteria should be identifiable from, and match the PICO question. Specifically, studies selected for inclusion are not based on clearly defined criteria. So inclusion and exclusion criteria must be specific.
R: The section included between rows 520-536 was edited to better reflect the required changes: “In this project, we included articles published in English probing the association of PGx tests with pharmacological treatment outcomes for SMI (i. e. BD, MDD, SCZ) and reporting on pharmacokinetic markers. We defined treatment outcomes as a response to the practiced treatment regimen, as reported by the authors. Accepted study designs included 1) open-label trials, 2) randomized controlled trials, 3) cross-sectional studies, 4) retrospective cohort studies, 5) prospective cohort studies, 6) recruiting human subjects ≥ 18 years old and assessing treatment outcomes as defined by the study authors. We excluded: 1) meta-analyses, 2) systematic reviews, 3) case reports, 4) case series, 5) letters to the editor and 6) editorials. No time restriction was applied based on the year of publication. Pharmacodynamic markers and studies assessing the safety or tolerability profile of pharmacological treatments have been excluded.”
4) Why does the PRISMA flow diagram end with (n=N/A). That box is redundant.
The authors give a clear view of the results in the tables, but without providing any numerical results (only descriptions). Also, the forest plot of effect sizes and the overall summary effect of studies were missed (here, I think of numerical results, not descriptive ones). There are no numerical results!!!??? Additionally, the forest plots, which need to summarize the findings, are missing. For example, forest plot estimates (e.g., mean) and confidence intervals (e.g., 95% CI) represented by whiskers for studies and/or multiple findings within a study in a horizontal orientation, are missed.
Specifically, the systematic review needs to be presented as the best form of evidence because it is positioned at the top of the hierarchy of evidence. Accordingly, the summary results (numbered results) are very important to see the main result and give the main conclusion of this study.
R) The box describing the selection process is reproduced as part of the standard PRISMA 2009 flowchart, but no quantitative meta-analytical analysis was performed for this project. To prevent misunderstandings, this has been removed from the original flowchart.
5) What are the limitations of this study?? The implications for practices or future research are not quite clear because a summary result from the forest plot is missing.
In view of the extensiveness of the research area and the extensiveness of the number of studies, the discussion is too short and deficient.
R) To better reflect the required changes an additional paragraph has been added 4. Limitations (right after conclusions and before the 5. Materials and methods). No meta-analysis has been performed for this systematic review paper, therefore we produced no forest plot. The Discussion has been expanded to reflect the reviewer’suggestions.
6) This systematic review does not give a clear conclusion on the clinical significance of the research question (which was not defined). Also, the heterogeneity of this systematic review is missing (in results), and that result should confirm whether the selected works are consistent with the research question, generate a concise conclusion to this study, and impact future studies. The conclusion does not clearly indicate the significance of improving pharmacological treatment selection in psychiatry.
R) The conclusion section has been edited to better focus on the papers included in the present project as follows: “The results of our review seem to point to a significant heterogeneity in assessed outcomes and in the testing panels. Only three papers included in the present project reported on PGx testing in BD, with only one RCT [29]. Considering the current relatively limited number of papers dedicated to the topic, the evidence regarding PGx testing for treatment selection in BD appears particularly scarce. In our data synthesis, less than half of the total studies dedicated to SCZ reported a positive association between PGx and treatment outcomes, and among them three focused on ABCB1 polymorphisms and three additional papers reported on CYP2D6 polymorphisms. The only RCT included in this project and dedicated to assessing PGx testing in SCZ was negative [27]. At this stage the evidence supporting the use of PGx testing in isolation to predict treatment outcomes in SCZ do not appear particularly poignant. Blood drug monitoring may represent an additional resource in guiding pharmacological treatment dosing, with clinical practice guidelines specifically dedicated to optimize their use[79]. Arguably, PGx testing may be synergistically integrated with psychotropic blood monitoring to fully exploit these two different sources of information in optimizing the therapeutic and safety profile for each medication trial. Our study selection did not include any study employing combinatorial pharmacogenomic testing for predicting pharmacological treatment outcomes in SCZ. Fourteen of the twenty-three included studies focusing on MDD described a positive association between PGx testing and pharmacological treatment outcomes [28, 30, 31, 33, 34, 36, 39, 56, 57, 60-62, 65, 66]. Five of the ten RCT dedicated to MDD described a positive association for PGx testing and treatment outcomes [28, 31, 35, 36, 39], but considering the significant heterogeneity in the testing panels involved, no firm conclusion can be reasonably drawn from our results.”
Reviewer 3 Report
The review by Paribello et al. is well-written and comprehensive. I would recommend the following changes before accepting it for publication.
1. Was any data mining tools or programs used for the study?
2. Regarding the ages of individuals part of the studies in Table1,2 and 3, what’s the youngest and oldest age? Please also mention in the manuscript.
3. Table1, 2, and 3 should be also made available in a more interactive format, such as on GitHub so that its easier for users to navigate and extract information.
4. The main input set (with 1457 records) should be made available on GitHub.
5. Are there repetitions in terms of studies between the three tables? Which ones?
6. Nothing is mentioned about the last block in Figure 1, “studies included in quantitative synthesis, n=NA). Was this performed and then mentioned here?
Author Response
The review by Paribello et al. is well-written and comprehensive. I would recommend the following changes before accepting it for publication.
1) Was any data mining tools or programs used for the study?
R) Rayyan a semi-automation tool, was employed to support the screening process. On row 424 of the present manuscript it is reported as follows: “Rayyan, a semi-automation tool, was employed to facilitate the screening process [87].”. We did not employ other mining tools for this project.
2) Regarding the ages of individuals part of the studies in Table 1, 2 and 3, what’s the youngest and oldest age? Please also mention in the manuscript.
R) Unfortunately, this piece of information is missing from several papers included in the qualitative analysis.
3) Table1, 2, and 3 should be also made available in a more interactive format, such as on GitHub so that its easier for users to navigate and extract information.
R) Table 1, 2, 3 are also available on GitHub at the following link https://github.com/claudiapis/tables_pharmacokinetic_markers
4) The main input set (with 1457 records) should be made available on GitHub.
R) The records screened for this project have been added to the following address https://github.com/pasqualeparibell/Pharmacokinetic-markers-of-clinical-outcomes-in-severe-mental-illness-a-systematic-review.---source/tree/main
5) Are there repetitions in terms of studies between the three tables? Which ones?
Seventeen papers are included in table 1, twenty-three in table 2 and three in table 3. In total, 43 records are described in all the three tables taken together for 42 papers included in the qualitative analysis. Among them, McCarthy et al., 2021 [29] is mentioned in both table 2 and table 3 as it recruited individuals affected by MDD and BD.
R) We confirm that the paper of McCarthy et al. reported findings from a combined sample of patients affected by BD and MDD. This has been specified in the revised results section: “An additional paper [29] reported the association of CGI changes with PGx testing of a mixed population comprising BD, PTSD and MDD.”
6) Nothing is mentioned about the last block in Figure 1, “studies included in quantitative synthesis, n=NA). Was this performed and then mentioned here?
R) The box is reproduced as part of the standard PRISMA 2009 flowchart, but no quantitative analysis was performed. To prevent misunderstandings, this has been removed from the original flowchart.
Reviewer 4 Report
The review is well-written and covers the work done in this area. In the abstract, Please summarize the recent findings about the pharmacokinetic markers in severe mental illness and separate the introduction into paragraphs to facilitate the reading.
Thank you,
Author Response
1) The review is well-written and covers the work done in this area. In the abstract, Please summarize the recent findings about the pharmacokinetic markers in severe mental illness and separate the introduction into paragraphs to facilitate the reading.
R) We thank the reviewer for the positive assessment of our work. The introduction section has been edited as suggested also by reviewer #1 to a more condensed version. The full table is available as SI.
Round 2
Reviewer 1 Report
I appreciate the extensive work in revising the manuscript and commend the authors on their overall effort and contribution.
Reviewer 2 Report
The authors significantly improved the article. Although I do not absolutely agree with some parts of the article, the authors have explained their positions. According to that, considering the explanations they gave in their answers and the additionally refined text, I have no additionally comments.